# Mixing times and cutoffs in open quadratic fermionic systems

E. Vernier

CNRS & LPSM, Université Paris Diderot, place Aurélie Nemours, 75013 Paris, France.
vernier@lpsm.paris

April 28, 2020

## Abstract

In classical probability theory, the term *cutoff* describes the property of some Markov chains to jump from (close to) their initial configuration to (close to) completely mixed in a very narrow window of time. We investigate how coherent quantum evolution affects the mixing properties in two fermionic quantum models (the "gain/loss" and "topological" models), whose time evolution is governed by a Lindblad equation quadratic in fermionic operators, allowing for a straightforward exact solution. We check that the cutoff phenomenon extends to the quantum case and examine how the mixing properties depend on the initial state. In the topological case, we further show how the mixing properties are affected by the presence of a long-lived edge zero mode when taking open boundary conditions.

# 1   Introduction

Coupling to an external environment is one of the many ways to drive a classical or quantum system out of equilibrium. Besides its relevance in experiments or realistic materials, where the influence of the environment can very rarely be discarded, it also offers the possibility of creating new types of steady states, and has thereby received an increase of interest over the last decade [1–3]. A particularly important parameter is then the time needed for the system to reach equilibrium, which, depending on the situation, one may want to be as long (e.g., in quantum memory devices) or as short as possible (e.g., in Monte-Carlo samplings of the steady state) [4]. When the system's effective dynamics can be treated as Markovian, which occurs whenever the environment relaxes much faster than the system and keeps no memory of its interaction with the latter, relaxation towards the steady state is generally characterized by the relaxation time $t_{\rm rel}$, obtained as the inverse of the spectral gap of the Liouvillian (in the quantum case [5,6]) or the transition matrix (in the classical case [10]).

   One should however bear in mind that, while the spectral gap gives information about the late time convergence of physical observables, this information may often not be enough to quantify how far the system is from equilibrated (or "mixed") at a given time $t$. In other terms, while the gap informs us of an exponential decay of the form $Ce^{-\bar{\lambda}t}$, it does not tell anything about the prefactor $C$, nor about the actual behaviour of the system at times which are not $\gg t_{\rm rel}$ [11]. This fact has been at the source of a widespread interest among the mathematical community, and the study of the mixing times of classical Markov chains has expanded over the last thirty years as one of the most active branches of probability theory [10, 11]. A remarkable result in this field, observed in many Markov chains when some parameter such as the size, number of particles or dimensionality is growing large, is the emergence of a *cutoff phenomenon*, a sharp transition which sees the system reach equilibrium over a narrow window of time [7–11]. The historical example was the shuffling of a deck of 52 cards, where it was shown that 7 shuffles are enough to bring the distribution to close to random, whereas less shuffles still retain a strong memory of the intial ordering and more than 7 shuffles do not significantly alter the mixing [7,8]. However, cutoffs have kept on attracting attention ever since and were proven to appear in a number of classic situations, including random walks on hypercubes [12], simple one-dimensional exclusion processes [13,14] or the Glauber dynamics of statistical models such as the two-dimensional Ising model in its high-temperature phase [15].

   It seems natural at this stage to ask whether an equivalent phenomenon exists in the quantum context. This was in fact already adressed in [16], where an appropriate distance to equilibrium was defined in the quantum information language and the existence of cutoff

established in some specific cases. However these results remain tied to some restrictions on the types of systems considered as well as on the nature of the initial states, and leave several open questions, for instance relating to the initial state dependence of the mixing properties.

Rather than generic theorems, our focus in this work will be to study mixing properties in a paradigmatic example of open quantum system, consisting in a free fermionic Hamiltonian linearly coupled to an external bath. More specifically, there are two types of system-bath coupling we shall consider : one corresponds to gain/loss of particles through interaction with the environment, and the other may be considered as a toy-model for Liouvillians with non-trivial topological properties [17]. Both these couplings have in common that they reduce in the classical limit to the master equation for the hypercube random walk, and are therefore good candidates for studying the interplay of quantum coherence and classical cutoffs. Another advantage of such models is that they can be solved exactly using free-fermion techniques [18], and can therefore be used to make analytical predictions on the mixing properties (in [19] free-fermionic chains linearly coupled to a bath were already studied in this perspective, generic bounds on the mixing times were provided, but no discussion of a cutoff phenomenon was made).

Our work is organized as follows. In Section 2, we introduce the models and review their classical limit. In Section 3, we describe the diagonalization of the Liouvillians in terms of a complete set of master modes. We also put forward the topological features of one of our models, in particular the existence of an edge zero mode when open boundary conditions are taken. In Section 4 we turn to the mixing properties, and construct in terms of the master modes a family of *factorized* initial states, from which the exact time evolution may be computed, and which we argue from numerics that they are good representatives of the "worst" and "best" conditions for mixing among all possible initial states, that is, those which lead at a given time $t$ to the least or most possible mixing. From there, we conclude in Section 5 about the existence of a cutoff in all cases (defined, as customarily, from the worst mixing state at all times), and further describe the dependence of the mixing time on the choice of initial state. An interesting exception, discussed in Section 5.4, is the case of the "topological" model with open boundary conditions, where the existence of the zero mode results in a destruction of the cutoff, a phenomenon we do not know of an analog in classical problems. We also discuss in Section 5.3 the relation with other physical quantities, in particular the von Neumann entropy and local observables. Our findings are summarized in Section 6.

## 2 The models

### 2.1 Fermionic chains with linear dissipation

We consider the evolution of an open quantum system, that is a quantum system coupled to an external environment. In the Markovian description, where the coupling is supposed to be weak enough, and the environment's dynamics fast enough such that the latter does not keep any memory of its interaction with the system, the dynamics of the system's density matrix is known to be well described by an equation of the Lindblad form [6]

$$\frac{\mathrm{d}\rho}{\mathrm{d}t} = \mathcal{L}\rho := -i[H, \rho] + \mathcal{L}_{\mathrm{D}}\rho, \qquad \mathcal{L}_{\mathrm{D}}\rho = \sum_{\mu} \left( L_{\mu}\rho L_{\mu}^{\dagger} - \frac{1}{2}\{L_{\mu}^{\dagger}L_{\mu}, \rho\} \right), \qquad (1)$$

where the Liouvillian $\mathcal{L}$ is formed of a Hamiltonian part accounting for the system's unitary evolution, and a part $\mathcal{L}_{\mathrm{D}}$ describing the coupling with the environment. [ , ] and

$\{,\}$ are respectively the commutator and anticommutator, $[A, B] := AB - BA$, $\{A, B\} := AB + BA$.

The Hamiltonian is taken here to be that of a free fermionic chain,

$$H = -g \sum_{j=1}^{L} \left[ c_j^\dagger c_{j+1} + c_{j+1}^\dagger c_j + \alpha \left( c_j^\dagger c_{j+1}^\dagger + c_{j+1} c_j \right) \right] - gh \sum_{j=1}^{L} c_j^\dagger c_j \,, \qquad (2)$$

where the $c_j^\dagger, c_j$ are fermionic creation/annihilation operators, satsifying the canonical anticommutation rules $\{c_i^\dagger, c_j\} = \delta_{i,j}$, $\{c_i, c_j\} = \{c_i^\dagger, c_j^\dagger\} = 0$. Through a Jordan-Wigner transformation [20, 21],

$$c_j + c_j^\dagger = \left( \prod_{l<j} \sigma_l^z \right) \sigma_j^x \,, \qquad i(c_j - c_j^\dagger) = \left( \prod_{l<j} \sigma_l^z \right) \sigma_j^y \,, \qquad (-1)^{\mathcal{Q}} = \prod_{l=1}^{L} \sigma_l^z \,, \qquad (3)$$

the Hamiltonian (2) can also be rewritten as that of a XY spin-1/2 chain with transverse magnetic field, namely

$$H = g \left( \sum_{j=1}^{L-1} \left[ \frac{1+\alpha}{2} \sigma_j^x \sigma_{j+1}^x + \frac{1-\alpha}{2} \sigma_j^y \sigma_{j+1}^y \right] - (-1)^{\mathcal{Q}} \left[ \frac{1+\alpha}{2} \sigma_L^x \sigma_1^x + \frac{1-\alpha}{2} \sigma_L^y \sigma_1^y \right] \right) - \frac{gh}{2} \sum_{j=1}^{L} \left( \sigma_j^z + 1 \right) \,,$$
$$(4)$$

where the matrices $\sigma_j^{x,y,z}$ act as Pauli matrices on the $j^{\text{th}}$ site of the chain, and as identity elsewhere. The case $\alpha = 1$, in particular, corresponds to the Ising chain in a transverse magnetic field, or, expressed in terms of Majorana fermions $w_{2j-1} = c_j + c_j^\dagger$, $w_{2j} = i(c_j - c_j^\dagger)$, the Kitaev chain [22].

As for the dissipative part $\mathcal{L}_{\text{D}}$, we will consider in this work two choices of Lindblad jump operators $L_\mu$, both linear in the fermions. As a result the Lindblad equation (1) is quadratic, and can be diagonalized exactly [18]. The first case we will consider, dubbed "gain/loss" in the following, corresponds to two types of jump operators for each site of the chain,

$$L_j^{\text{gain}} = \sqrt{\gamma} c_j^\dagger \,, \qquad L_j^{\text{loss}} = \sqrt{\gamma} c_j \,, \qquad (5)$$

corresponding to gain and loss of fermions through interaction with the environment. Such models have been considered in many places in the past literature, see for instance [23, 24].

Another case we will consider, dubbed "topological" for reasons that will be explained below (see Section 3.3), corresponds to the following choice of jump operators on each site :

$$L_j^{\text{top}} = \sqrt{\gamma}(c_j + c_j^\dagger) \,. \qquad (6)$$

Contrarily to the gain/loss case, it is not clear how to realize the above operators in a realistic physical setting. Nevertheless, we will study them as a particularly simple toy model for dissipative fermionic systems exhibiting topological features, such as those considered in [17, 25, 26] (we note in particular that our model corresponds to a particular choice of the model considered in [26], namely $\Delta = 1$).

## 2.2 The classical part, and the cutoff phenomenon

A reason for the choices (5), (6) of Lindblad operators is that both lead, in the purely dissipative limit (that is, when the Hamiltonian part is removed from (1)), to a well-known

classical Markovian problem. The latter is obtained by restricting to density matrices diagonal in the basis of fermion occupation numbers, whose diagonal entries we label as $\rho_{n_1,\dots n_L}$, $n_i \in \{0,1\}$. As can easily be checked, both gain/loss and topological models lead to the same master equation

$$\frac{\mathrm{d}}{\mathrm{d}t}\rho_{n_1,\dots n_L} = \gamma \sum_{j=1}^{L} \left(\rho_{n_1\dots\bar{n}_j\dots n_L} - \rho_{n_1\dots n_j\dots n_L}\right), \qquad \bar{n}_j := 1 - n_j, \tag{7}$$

which is the master equation for a classical nearest-neighbour random walk on the $L-$dimensional hypercube $\{0,1\}^L$ (the latter is also equivalent to the Glauber dynamics of the classical Ising model at infinite temperature [15]). The walk, whose position is indexed by the $L$-uple $n_1,\dots n_L$, performs nearest neighbour jumps of rate $L\gamma$, corresponding to each of its components changing value at rate $\gamma$. As $t \to \infty$, it reaches the uniform stationary state, where all the $\rho_{n_1,\dots n_L}$ are equal to $1/2^L$.

Starting from an initial configuration $\rho(0)$ (corresponding to a set of non-negative densities $\rho_{n_1,\dots n_L}(0)$ normalized to $\sum_{\{n_i\}} \rho_{n_1,\dots n_L} = 1$), a good way to quantify how fast equilibration occurs is through the total variation distance to equilibrium [10, 11]

$$||\rho(t) - \rho(\infty)|| = \frac{1}{2} \sum_{n_1,\dots n_L \in \{0,1\}} |\rho_{n_1,\dots n_L}(t) - \rho_{n_1,\dots n_L}(\infty)|. \tag{8}$$

One way to think of the total variation distance between two distributions is as the maximum difference between probabilities associated to a single event. The total variation distance to equilibrium has several well-known properties which hold for any Markov chain [11], in particular it is always comprised between 0 and 1, and is non-increasing with time.

Given (8), a very important quantity is

$$d(t) = \max_{\rho(0)} ||\rho(t) - \rho(\infty)||, \tag{9}$$

which is at a given time the maximal distance to equilibrium over all possible initial configurations. Looking back at the particular case of the random walk on $\{0,1\}^L$, the maximal distance is obtained at any time by taking $\rho(0)$ to be any purely localized state, where one of the $\rho_{n_1,\dots n_L}(0)$ equals 1 and the others equal 0 [12]. As can be seen on Figure 1, and reviewed in more detail in Appendix A, the distance jumps from 1 to 0 around a time $t_{\mathrm{mix}}(L) = \frac{\ln L}{2\gamma}$, and this jump occurs in a time window of width $O(1)$, which becomes much smaller than $t_{\mathrm{mix}}$ as $L$ increases: this characterizes what has been coined a cutoff by the mathematical community [7, 9, 10]. More generally, a sequence of Markov chains indexed by some size or dimensionality $L$ are said to exhibit a cutoff if there exist a sequence of mixing times $t_{\mathrm{mix}}(L)$ (typically increasing with $L$) such that for any $\epsilon > 0$, one has [11]

$$d\left((1-\epsilon)t_{\mathrm{mix}}(L)\right) \xrightarrow[L\to\infty]{} 1 \tag{10}$$

$$d\left((1+\epsilon)t_{\mathrm{mix}}(L)\right) \xrightarrow[L\to\infty]{} 0. \tag{11}$$

## 3 Diagonalization of the Liouvillians

### 3.1 The gain/loss model

Lindblad equations of the form (1), quadratic in fermion operators, have been presented and diagonalized in [18]. While generically the diagonalization goes through re-expressing

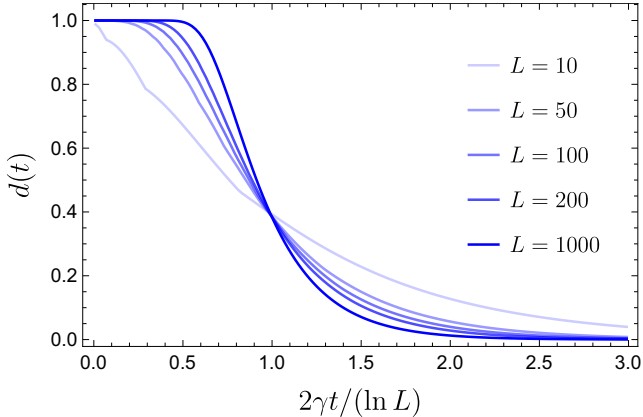

Figure 1: Distance to equilibrium of the classical random walk on the hypercube $\{0,1\}^L$. As $L$ increases, a sharp cutoff develops, with the distance jumping from 1 to 0 around a time $t_{\text{mix}} = \frac{\ln L}{2\gamma}$.

the Liouvillian as a quadratic "Hamiltonian" acting on a superspace of operators and reduces the diagonalization of the latter to that of a $4L$-dimensional matrix, in the present case translation invariance makes it natural to reduce the problem to a block-diagonal form without stepping to the super-operator formalism. We therefore introduce the momentum space creation and annihilation operators

$$c_k = \frac{1}{\sqrt{L}} \sum_{j=1}^{L} e^{ikj} c_j, \qquad k \in \{k_0, k_1, \ldots k_{L-1}\} := \left\{ 0, \frac{2\pi}{L}, \ldots \frac{2\pi(L-1)}{L} \right\}, \qquad (12)$$

and it shall be clear depending on the context whether we are using real space or momentum space operators (we will try as much as possible to reserve the letter $j$ for the sites of the chain and $k$ for the momenta).

A first step is the diagonalization of the Hamiltonian (2). This is achieved by introducing the Bogoliubov-rotated fermions [21],

$$\eta_k = \begin{cases} e^{-i\frac{\pi}{4}} c_k^\dagger & k = 0 \\ e^{i\frac{\pi}{4}} c_k & k = \pi \\ e^{-i\frac{\pi}{4}} \cos\theta_k c_k^\dagger - e^{i\frac{\pi}{4}} \sin\theta_k c_{-k} & \text{otherwise} \end{cases} , \qquad \tan(2\theta_k) = \frac{\alpha \sin k}{\frac{h}{2} + \cos k}. \qquad (13)$$

which satisfy the canonical anticommutation relations $\{\eta_k, \eta_{k'}\} = \{\eta_k^\dagger, \eta_{k'}^\dagger\} = 0$, $\{\eta_k, \eta_{k'}^\dagger\} = \delta_{k,k'}$, and in terms of which the Hamiltonian (2) becomes

$$H = \sum_k \epsilon_k \left( \eta_k^\dagger \eta_k - \frac{1}{2} \right) - \frac{ghL}{2}, \qquad \epsilon_k = 2g\sqrt{\left( \cos k + \frac{h}{2} \right)^2 + \alpha^2 \sin^2 k}. \qquad (14)$$

Turning to the dissipative part $\mathcal{L}_D$, it is easy to check that we can rewrite it for the gain-loss model as

$$\mathcal{L}_D \rho = \gamma \sum_k \left( c_k^\dagger \rho c_k + c_k \rho c_k^\dagger - \rho \right) = \gamma \sum_k \left( \eta_k^\dagger \rho \eta_k + \eta_k \rho \eta_k^\dagger - \rho \right). \qquad (15)$$

Gathering (14) and (15), we can now compute the action of the full Liouvillian $\mathcal{L}$ on the rotated fermions $\eta_k, \eta_k^\dagger$. Because of their anticommuting nature, these are not simply annihilated by the components of $\mathcal{L}$ with momentum $k' \neq k$. Therefore, it will turn convenient to introduce the modified fermions

$$\bar{\eta}_k = \eta_k(-1)^{\mathcal{Q}} \qquad \bar{\eta}_k^\dagger = (-1)^{\mathcal{Q}}\eta_k^\dagger = -\eta_k^\dagger(-1)^{\mathcal{Q}}, \tag{16}$$

so that $[\bar{\eta}_k^{(\dagger)}, \eta_{k'}^{(\dagger)}] = 0$ for $k \neq k'$. We can check from there

$$\mathcal{L}\,\mathrm{id} = 0 \tag{17}$$

$$\mathcal{L}\bar{\eta}_k^\dagger = (-\gamma - i\epsilon_k)\bar{\eta}_k^\dagger := -2\beta_k^+ \bar{\eta}_k^\dagger \tag{18}$$

$$\mathcal{L}\bar{\eta}_k = (-\gamma + i\epsilon_k)\bar{\eta}_k := -2\beta_k^- \bar{\eta}_k \tag{19}$$

$$\mathcal{L}[\bar{\eta}_k^\dagger, \bar{\eta}_k] = -2\gamma[\bar{\eta}_k^\dagger, \bar{\eta}_k] = -2(\beta_k^+ + \beta_k^-)[\bar{\eta}_k^\dagger, \bar{\eta}_k], \tag{20}$$

and therefore the modified fermions $\bar{\eta}_k, \bar{\eta}_k^\dagger$ can be used to define a complete set of eigenmodes (*master modes*) of the Liouvillian. These are indexed by a sequence of $2L$ binary integers $\underline{\nu} = (\nu_0^+, \nu_0^-, \nu_1^+, \nu_1^-, \dots \nu_{L-1}^+, \nu_{L-1}^-)$, $\nu_i^\pm \in \{0, 1\}$, and read

$$\mathcal{C}_{\underline{\nu}} = \prod_{i=0}^{L-1} [(\bar{\eta}_{k_i})^{\nu_i^+} (\bar{\eta}_{k_i}^\dagger)^{\nu_i^-}] \tag{21}$$

where the bracket notation indicates that if for a given $i$ both $\nu_i^\pm$ are $= 1$, the factor $[\bar{\eta}_{k_i}\bar{\eta}_{k_i}^\dagger]$ should be understood as the commutator $[\bar{\eta}_{k_i}, \bar{\eta}_{k_i}^\dagger]$. The associated eigenvalues of the Liouvillian, namely $\mathcal{L}\mathcal{C}_{\underline{\nu}} = \lambda_{\underline{\nu}}\mathcal{C}_{\underline{\nu}}$, read

$$\lambda_{\underline{\nu}} = -2\sum_{i=0}^{L-1}(\nu_i^+ \beta_{k_i}^+ + \nu_i^- \beta_{k_i}^-). \tag{22}$$

Since all $\beta_k^\pm$ have a positive real part, the identity (or, rather, $\rho_\infty := \frac{1}{2^L}\mathrm{id}$), is the only mode with eigenvalue 0 and therefore corresponds to the unique steady state. Relaxation towards the steady state occurs exponentially at late times, with a rate given by the eigenvalue with smallest real part (in absolute value), the so-called *spectral gap* $\bar{\lambda} = \gamma$. We accordingly define the *relaxation time* as the inverse of the spectral gap,

$$t_{\mathrm{rel}} = 1/\bar{\lambda} = \frac{1}{\gamma}. \tag{23}$$

## 3.2 The topological model

We now turn to the topological model (6). The Hamiltonian part is the same as for the gain/loss model, so we look directly at the action of the dissipative part. In terms of the rotated fermions (13), we check

$$\mathcal{L}_{\mathrm{D}}\rho = \gamma\sum_{j=1}^{L}\left((c_j^\dagger + c_j)\rho(c_j^\dagger + c_j) - \rho\right) = \gamma\sum_{k}\left((\eta_k^\dagger + \eta_{-k})\rho(\eta_k + \eta_{-k}^\dagger) - \rho\right), \tag{24}$$

which leads to the following action of the full Liouvillian

$$\mathcal{L}\bar{\eta}_k = (-\gamma + i\epsilon_k)\bar{\eta}_k - i\gamma\bar{\eta}_{-k}^\dagger \tag{25}$$

$$\mathcal{L}\bar{\eta}_k^\dagger = (-\gamma - i\epsilon_k)\bar{\eta}_k^\dagger + i\gamma\bar{\eta}_{-k}. \tag{26}$$

Let us define new fermion operators

$$\Gamma_k = \frac{e^{i\frac{\pi}{4}}\eta_k + e^{-i\frac{\pi}{4}}\eta_{-k}}{\sqrt{2}}, \qquad \Gamma_k^\dagger = \frac{e^{-i\frac{\pi}{4}}\eta_k^\dagger + e^{i\frac{\pi}{4}}\eta_{-k}^\dagger}{\sqrt{2}}, \tag{27}$$

and similarly $\bar{\Gamma}_k = \Gamma_k(-1)^{\mathcal{Q}}$, $\bar{\Gamma}_k^\dagger = (-1)^{\mathcal{Q}}\Gamma_k^\dagger$. $\Gamma_k$ and $\Gamma_k^\dagger$ satisfy the same canonical anticommutation relations as the fermions $\eta_k$, $\eta_k^\dagger$. The eigenmodes of the Liouvillian can be constructed from (25), (26), and these are conveniently reexpressed in terms of (27) as

$$\mathcal{C}_k^+ = \left(\frac{\epsilon_k - \sqrt{\epsilon_k^2 - \gamma^2}}{\epsilon_k + \sqrt{\epsilon_k^2 - \gamma^2}}\right)^{1/4} \bar{\Gamma}_k - \left(\frac{\epsilon_k - \sqrt{\epsilon_k^2 - \gamma^2}}{\epsilon_k + \sqrt{\epsilon_k^2 - \gamma^2}}\right)^{-1/4} \bar{\Gamma}_k^\dagger \tag{28}$$

$$\mathcal{C}_k^- = \left(\frac{\epsilon_k - \sqrt{\epsilon_k^2 - \gamma^2}}{\epsilon_k + \sqrt{\epsilon_k^2 - \gamma^2}}\right)^{-1/4} \bar{\Gamma}_k - \left(\frac{\epsilon_k - \sqrt{\epsilon_k^2 - \gamma^2}}{\epsilon_k + \sqrt{\epsilon_k^2 - \gamma^2}}\right)^{1/4} \bar{\Gamma}_k^\dagger \tag{29}$$

$$\mathcal{C}_k^\pm = [\bar{\Gamma}_k^\dagger, \bar{\Gamma}_k]. \tag{30}$$

One indeed checks

$$\mathcal{L} \, \text{id} = 0 \tag{31}$$

$$\mathcal{L}\mathcal{C}_k^+ = (-\gamma - i\sqrt{\epsilon_k^2 - \gamma^2})\mathcal{C}_k^+ := -2\beta_k^+ \mathcal{C}_k^+ \tag{32}$$

$$\mathcal{L}\mathcal{C}_k^- = (-\gamma + i\sqrt{\epsilon_k^2 - \gamma^2})\mathcal{C}_k^- := -2\beta_k^- \mathcal{C}_k^- \tag{33}$$

$$\mathcal{L}\mathcal{C}_k^{+-} = -2\gamma\mathcal{C}_k^{+-} = -2(\beta_k^+ + \beta_k^-)\mathcal{C}_k^{+-}, \tag{34}$$

and from there the master modes can be constructed as for the gain/loss model, with eigenvalues of the form (22). Depending on the value of $\gamma/g, \alpha, h$, the band structure of the eigenvalues $-2\beta_k^\pm$ may draw different regimes. This is illustrated on Figure 2, where for simplicity we have specialized to $\alpha = 1$, corresponding to the Ising chain in a transverse magnetic field. The expression of the spectral gap, indicated by the black arrow on the figure, depends on the regime under consideration. For $\alpha = 1$, it is given by

$$\bar{\lambda} = \begin{cases} \gamma & \text{if } \frac{\gamma}{g} \leq 2 - |h| \\ \gamma - \sqrt{\gamma^2 - g^2(2 - |h|)^2} & \text{if } \frac{\gamma}{g} \geq 2 - |h|. \end{cases} \tag{35}$$

In contrast to the gain-loss case, we note that the gap closes here for $g \to 0$, as a result of there being many other steady states in this limit.

## 3.3 What is topological about the topological model ?

It is well known that in the regime of parameters $|h| < 2$, $\alpha \neq 0$ the Hamiltonian (2) is in a topologically non-trivial phase, with a gapped bulk and gapless edge modes [22]. This is best seen in terms of the Majorana fermions $w_{2j-1} = c_j + c_j^\dagger$, $w_{2j} = i(c_j - c_j^\dagger)$, in the extreme limit $\alpha = 1$, $h = 0$: the Hamiltonian is then a sum of bilinears of the form $w_{2j}w_{2j+1}$, which in the case of open boundary conditions leaves the modes $w_1$ and $w_{2L}$ unpaired. These edge modes commute with the Hamiltonian, anticommute with the fermion number parity $(-1)^{\mathcal{Q}}$, and are therefore responsible for an exact degeneracy of the spectrum between sectors $(-1)^{\mathcal{Q}} = \pm 1$. Furthermore, these survive throughout the topological phase, where they can be expressed are a power series in $h$ [27]. They are then exponentially located at the edges, and the degeneracy holds exactly in the $L \to \infty$ limit.

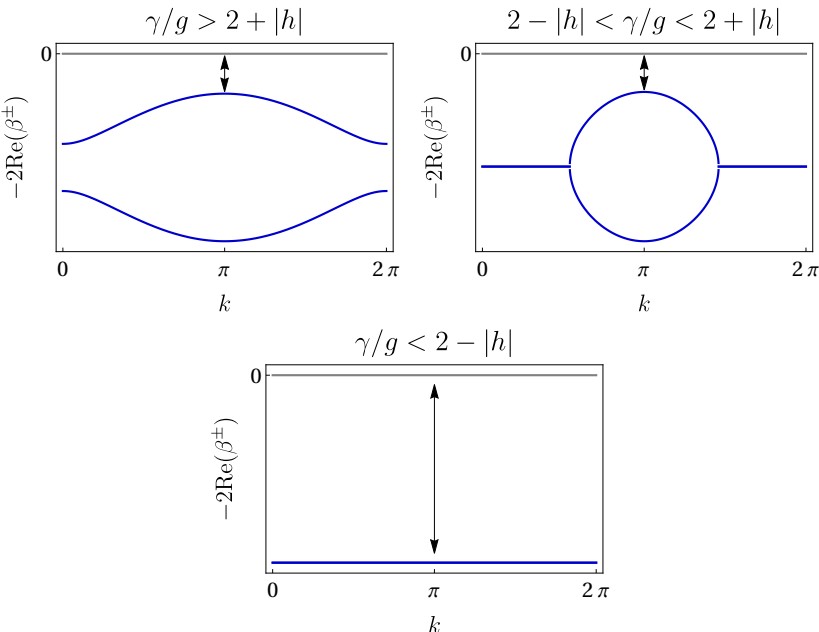

Figure 2: Band structure for the elementary excitations of the Liouvillian $\mathcal{L}$ for the topological model, with $\alpha = 1$. Depending on the relative values of $\gamma/g$ and $h$, there are three regimes : two bands (left), one band (right), and an intermediate regime (middle). The black arrow represents the spectral gap $\bar{\lambda}$ in each case.

We will now see that analogous features hold for the Liouvillian of the topological model, in the same regime of parameters. Recasting the action of the dissipative part $\mathcal{L}_\mathrm{D}$ in terms of spins, we see that it commutes with the operation

$$\Psi : \rho \to \sigma_L^x \rho\,, \tag{36}$$

Taking open boundary conditions for the Hamiltonian (2), $\sigma_L^x$ further commutes with $H$ in the limit $\alpha = 1$, $h = 0$, so the operation (36) commutes with the action of the full Liouvillian $\mathcal{L}$. Another important property of the operation (36) is that it anticommutes with the "parity of a-fermions" operator [18], which can be defined through its diagonal action on any product $\mathcal{C}$ of fermion operators as $\mathcal{Z}\mathcal{C} := (-1)^{\mathcal{Q}}\mathcal{C}(-1)^{\mathcal{Q}}$ (in other terms $\mathcal{Z}$ counts the parity of the number of fermion operators in the product $\mathcal{C}$). Since $\mathcal{Z}$ further commutes with the action of the Liouvillian, we see that $\Psi$ plays the role of a zero mode, and results for the Liouvillian in a doubly degenerate spectrum between sectors of parities $\mathcal{Z} = \pm 1$.

Switching to different values of $\alpha$ and $h$, we see that these features persist throughout an extended phase, namely whenever the Hamiltonian is in a nontrivial topological phase. More precisely, the degeneracies hold for $|h| < 2$, $\alpha \neq 0$, up to corrections exponentially small in the system size $L$. This is illustrated on Figure 3. In Section 5.4 we shall come back to these features, which will turn out to have interesting consequences for the mixing properties.

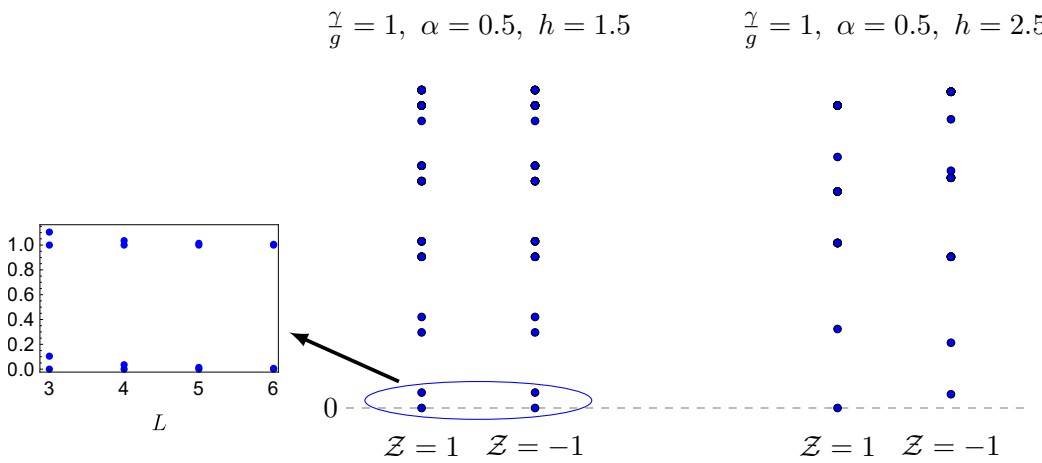

Figure 3: (Negated real part of the) spectrum of the Liouvillian for the topological model with open boundary conditions, split according to the value of the parity $\mathcal{Z} = \pm 1$, for values of the parameters inside the topological (left) or topologically trivial (right) phase. In the former case, the inset shows the first two eigenvalues in both sectors for increasing system sizes.

# 4 Mixing in the presence of quantum coherent evolution

Having at hand a complete set of master modes for both the gain/loss and topological models, we can now turn to the mixing properties of these models. In particular, we will be concerned with the fate of classical cutoff described in Section 2.2 when quantum coherent evolution is turned on ($g \neq 0$), and the dependence of the mixing properties on the choice of initial conditions. The notions of distance to equilibrium, mixing times and cutoffs have been extended to the quantum context in the past literature [16]. For a given quantum channel (Liouvillian) with a unique stationary state, the total variation distance (8) should be replaced by the trace distance

$$||\rho(t) - \rho_\infty|| = \frac{1}{2}\text{Tr}\sqrt{(\rho(t) - \rho_\infty)^\dagger(\rho(t) - \rho_\infty)}, \tag{37}$$

which, in the case where the matrices $\rho$ and $\rho_\infty$ are hermitian (as they will here), is simply half the sum of their absolute eigenvalues. The trace distance shares some important properties of the total varation distance, in particular it is comprised between 0 and 1, and non-increasing with time.

In this section we will examine the time-dependence of the distance (37) as a function of the initial configuration $\rho(0)$, with a particular interest in finding the "worst choice" initial conditions $\rho(0)$, which maximize the distance (37) at a given time $t$.

## 4.1 Preamble: a look at product chains

Both the gain/loss and topological Liouvillians split into $L$ momentum sectors, each carrying master modes $\mathcal{C}_k^+$, $\mathcal{C}_k^-$, $\mathcal{C}_k^{+-}$ which are eigenmodes of the Liouvillian and which commute or anticommute between different momentum sectors (for the topological model these were explicitly defined in equation (28)-(30), while for the gain/loss model they are simply the Bogoliubov fermions $\mathcal{C}_k^+ = \bar{\eta}_k^\dagger$, $\mathcal{C}_k^- = \bar{\eta}_k$, $\mathcal{C}_k^{+-} = [\bar{\eta}_k^\dagger, \bar{\eta}_k]$).

Let aside anticommutation, our models therefore resemble the so-called *product chains*, where the time evolution can be decomposed as the tensor product of $L$ independent

channels. In the classical setup, product chains (the hypercube random walk being an example) are known to generically exhibit a cutoff under mild assumptions [10, 28]. Some results were also established in the quantum context [16]: for instance, if the time evolution can be decomposed as a tensor product of identical quantum channels with a unique non-degenerate steady state, a cutoff was proven to hold when restricting to *separable* initial states, of the form $\rho(0) = \sum_i p_i \rho_1^{(i)} \otimes \ldots \otimes \rho_L^{(i)}$, $p_i \geq 0$, $\sum_i p_i = 1$. It is however not known in general, whether such states indeed maximize the distance (37) at all times. Another question raised in [16] is whether the worst choice of initial state at a given time $t$ is always a pure state, or whether, rather counterintuitively, states with some level of mixing might take slower to reach equilibrium (in the classical setup such a phenomenon has been presented in [32], where under the Glauber dynamics extra updates may result in delaying the mixing).

In order to illustrate these ideas, and before embarking into the study of the models defined in Section 2, let us briefly discuss a very simple example of product chain acting on $L$ independent qubits, where the Liouvillian acts on each qubit as : $\mathcal{L}\rho_k = ig[\sigma_k^z, \rho_k] + \gamma(\sigma_k^+ \rho_k \sigma_k^- + \sigma_k^- \rho_k \sigma_k^+ - \rho_k)$. One easily checks that $\mathcal{L}\sigma_k^0 = 0$, $\mathcal{L}\sigma_k^\pm = (-\gamma \pm ig)\sigma_k^\pm$, $\mathcal{L}\sigma_k^z = -2\gamma\sigma_k^z$, so here again there is a unique steady state $\rho_\infty = \frac{1}{2^L}\mathrm{id}$. For the sake of illustration, let us restrict to initial states of the product form, namely $\rho(0) = \rho_1(0) \otimes \ldots \otimes \rho_L(0)$. Each of the $\rho_k(0)$ can be decomposed as

$$\rho_k(0) = \frac{1}{2}\sigma_k^0 + \left(p_k - \frac{1}{2}\right)\left[\cos(2\theta_k)\sigma_k^z + \sin(2\theta_k)\left(e^{i\varphi_k}\sigma_k^+ + e^{-i\varphi_k}\sigma_k^-\right)\right] , \qquad (38)$$

where $0 \leq p_k \leq 1$, and therefore evolves as

$$\rho_k(t) = \frac{1}{2}\sigma_k^0 + \left(p_k - \frac{1}{2}\right)\left[e^{-2\gamma t}\cos(2\theta_k)\sigma_k^z + e^{-\gamma t}\sin(2\theta_k)\left(e^{i(\varphi_k+gt)}\sigma_k^+ + e^{-i(\varphi_k+gt)}\sigma_k^-\right)\right] . \qquad (39)$$

The eigenvalues of $\rho_k(0)$ are $p_k, 1 - p_k$, and therefore $p_k$ is a measure of the purity of the initial state : $p_k = 0$ or $1$ for pure quantum states, and $p_k = \frac{1}{2}$ for a completely mixed state. At time $t$ the eigenvalues of $\rho_k(t)$ take the form $\frac{1}{2} \pm (p_k - \frac{1}{2})f(\theta_k, t)$, and the distance to equilibrium is

$$||\rho(t) - \rho_\infty|| = \frac{1}{2}\sum_{\varepsilon_1 = \pm, \ldots \varepsilon_L = \pm}\left|\prod_{k=1}^{L}\left(\frac{1}{2} + \varepsilon_k\left(p_k - \frac{1}{2}\right)f(\theta_k, t)\right) - \frac{1}{2^L}\right| , \qquad (40)$$

where of course $\varepsilon_k = \pm$ should not be confused with the eigenenergies $\epsilon_k$ of (14). At any time this distance is maximized by maximizing (in absolute value) the terms $(p_k - \frac{1}{2})f(\theta_k, t)$ for each individual $k$. We observe here a peculiarity of two-level systems, namely that the purity of the initial states appears as a prefactor at all times, and, therefore, that the maximal distance is indeed obtained starting from a pure state [1]. More explicitly, it corresponds to choosing for each $k$ $p_k = 1$, and $\theta_k = \frac{\pi}{4}$, so the eigenvalues of $\rho_k(t)$ are $\frac{1}{2} \pm \frac{1}{2}e^{-\gamma t}$. The distance to equilibrium as a function of time then has the same form as that of the classical random walk on the hypercube discussed in Section 2.2 and Appendix A, and is indeed seen to exhibit a cutoff.

---

[1]Another way to arrive at the same conclusion would be to use the monotonicity of the distance, namely the fact that it can only decrease under the application of quantum channels [16]. Starting with $\rho(0)$ a product of density matrices with eigenvalues $p_k, 1 - p_k$, one can act on each qubit with negative time exponentials $e^{-t_k \mathcal{L}_k}$ in order to arrive at a pure quantum state, $\tilde{\rho}(0)$. By monotonicity arguments, it is then easy to see that $||e^{\mathcal{L}t}\tilde{\rho}(0) - \rho_\infty|| \geq ||e^{\mathcal{L}t}\rho(0) - \rho_\infty||$. Transposed to a product of internal spaces with larger dimension $d$, this argument only tells us that the maximal distance is obtained for initial density matrices which have "a certain level of purity", namely at least one zero eigenvalue in each tensorand.

## 4.2 Constructing initial states

We now move back to our models, which in contrast with product chains cannot be written as a tensor product of independent channels, due to the anticommutation between the operators $\mathcal{C}_k^\pm$ in different sectors To get an idea of the difficulties raised by the fermionic nature of the problem, let us look at the gain/loss model, starting with a single momentum sector. A generic initial density matrix can be written in the form (38), where $\sigma_k^0, \sigma_k^+, \sigma_k^-, \sigma_k^z$ are now replaced by id, $\bar{\eta}_k^\dagger, \bar{\eta}_k$ and $[\bar{\eta}_k^\dagger, \bar{\eta}_k]$. From (17)-(20) it is easy to read off its time evolution, and eigenvalues. As in the toy-model discussed above, the slowest mixing (namely, the maximal distance) is obtained by choosing $p_k = 1$ (or 0) and $\theta_k = \frac{\pi}{4}$. However, putting all momentum sectors back together, it is not anymore a valid choice to simply consider a product of such density matrices: contrarily to the case of product chains these do not commute with one another and therefore their product, being non-hermitian, does not correspond to a physical initial configuration. Similar remarks can be paralleled for the topological model.

In the following we will construct several classes of physical initial density matrices in terms of the master modes $\mathcal{C}_k^+$, $\mathcal{C}_k^-$, $\mathcal{C}_k^\pm$, which will turn out to be relevant for both the gain/loss and topological models (we recall that for the former the correspondence is $\mathcal{C}_k^+ = \bar{\eta}_k^\dagger$, $\mathcal{C}_k^- = \bar{\eta}_k$, $\mathcal{C}_k^{+-} = [\bar{\eta}_k^\dagger, \bar{\eta}_k]$).

**Single-sector commuting density matrices** A natural way to work around the problem of anticommutation is to start from products of single sector commuting density matrices, namely linear combinations of id and $\mathcal{C}_k^{+-}$ in each sector :

$$\rho_k^C(0) = \frac{1}{2}\mathrm{id} + \left(p_k - \frac{1}{2}\right)\mathcal{C}_k^{+-}, \tag{41}$$

with $0 \le p_k \le 1$. For both the gain/loss and topological models, these evolve in time as

$$\rho_k^C(t) = \frac{1}{2}\mathrm{id} + e^{-2\gamma t}\left(p_k - \frac{1}{2}\right)\mathcal{C}_k^{+-}, \tag{42}$$

and we write the corresponding single-sector eigenvalues

$$p_k^{C(1,2)}(t) = \frac{1}{2} \pm \left(p_k - \frac{1}{2}\right)e^{-2\gamma t}. \tag{43}$$

**Paired commuting density matrices** Another possibility is to combine the anticommuting master modes of different sectors into commuting objects. There are many ways to do this, but in the following we will restrict to the so-called paired density matrices built from two sectors with momenta $k_1, k_2$. We therefore define

$$\rho_{k_1,k_2}^C(0) = \frac{1}{4}\mathrm{id} + \frac{1}{2}\left(e^{i\varphi_{k_1,k_2}}\bar{\Gamma}_{k_1}^\dagger\bar{\Gamma}_{k_2}^\dagger + e^{-i\varphi_{k_1,k_2}}\bar{\Gamma}_{k_2}\bar{\Gamma}_{k_1}\right) + \frac{1}{4}[\bar{\Gamma}_{k_1}^\dagger, \bar{\Gamma}_{k_1}] \cdot [\bar{\Gamma}_{k_2}^\dagger, \bar{\Gamma}_{k_2}], \tag{44}$$

where $\bar{\Gamma}_k, \bar{\Gamma}_k^\dagger$ were defined in (27) for the topological model, while for the gain/loss model they should just be taken to be $\bar{\eta}_k, \bar{\eta}_k^\dagger$. Eq. (44) is the most general choice of a commuting combination corresponding to a pure state, that is with eigenvalues $(1, 0, 0, 0)$ (times the identity in other sectors). We will not justify here that starting from a pure state in paired sectors is indeed what maximizes the distance to equilibrium, but numerical studies below will confirm this fact. The explicit time dependence of $\rho_{k_1,k_2}^C(t)$ will be worked out separately for the gain/loss and the topological models in the next sections, and we will generically write the corresponding eigenvalues as

$$p_{k_1,k_2}^{C(1)}(t),\ p_{k_1,k_2}^{C(2)}(t),\ p_{k_1,k_2}^{C(3)}(t),\ p_{k_1,k_2}^{C(4)}(t). \tag{45}$$

**Density matrices for the full system**   The single-sector and paired commuting density matrices can now be combined into density matrices for the full system, namely arbitrary products of them can be taken. Furthermore, we may still multiply such products by one noncommuting single-sector density matrix $\rho_k(t)$. We therefore define

$$\rho_{\{(k)\},\{(k_{i_1},k_{j_1}),...(k_{i_m},k_{j_m})\},\{k_{l_1},...k_{l_n}\}}(t) := (\rho_k(t)) \prod_{a=1}^{m} \rho_{k_{i_a},k_{j_a}}^{\mathrm{C}}(t) \prod_{b=1}^{n} \rho_{k_{l_b}}^{\mathrm{C}}(t) \,, \qquad (46)$$

where the parentheses around $k$ in the left-hand side, and around $\rho_k(t)$ in the right-hand side, mean that the non-commuting density matrix $\rho_k(t)$ may or may not be present in the product. We must then have $2m + n(+1) = L$. In each sector (or pairs of sectors) the initial density matrices have internal parameters $(p_k, \varphi_k, \varphi_{k_1,k_2}, \text{etc...})$, which we leave unspecified for the moment. Writing the eigenvalues of $\rho_k(t)$ as $p_k^{(1,2)}(t)$, and those of the single-sector/paired commuting matrices as (43) and (45) respectively, we obtain the eigenvalues of (46) as products of the latter, which results in the following expression for the trace-norm distance to equilibrium

$$||\rho(t) - \rho_\infty|| \;=\; \frac{1}{2} \sum_{\substack{(c\in\{1,2\}) \\ d_1,...d_m\in\{1,2,3,4\} \\ e_1,...e_n\in\{1,2\}}} \left| \left( p_k^{(c)}(t) \right) \prod_{a=1}^{m} p_{k_{i_a},k_{j_a}}^{\mathrm{C}(d_a)}(t) \prod_{b=1}^{n} p_{k_{l_b}}^{\mathrm{C}(e_b)}(t) - \frac{1}{2^L} \right| \,, \qquad (47)$$

where once again parentheses account for the presence or absence of the non-commuting matrix $\rho_k(t)$.

Let us emphasize that the the density matrices (46) are only a very small subset of all the possible initial states. In particular, these are completely (or almost completely, because of the pairing between sectors) factorized. In the following, we will however observe from numerics that such density matrices still encompass the worst-choice initial state at any time.

## 4.3   The gain/loss model

We are now ready to examine the mixing properties of our models, starting with the gain/loss model. Before turning to numerics, we want to find the conditions which maximize the distance at any time $t$ for density matrices of the form (46). As can easily be checked, the distance (47) is generically maximized by separately optimizing the individual eigenvalues in each sector, that is taking the $p_k^{(c)}(t)$, $p_{k_{i_a},k_{j_a}}^{\mathrm{C}(d_a)}(t)$ and $p_{k_{i_b}}^{\mathrm{C}(e_b)}$ as close as possible to 0 or 1. Let us therefore see how this goes sector by sector.

**Single-sector commuting density matrices**   The time evolution of single-sector commuting matrices has been discussed in the previous section, and the associated eigenvalues found to be given by (43). Their contribution to the distance is maximized by taking $p_k = 1$ (or 0), that is, starting from a pure state in the corresponding sectors. Therefore,

$$p_k^{\mathrm{C}(1,2)}(t) = \frac{1}{2} \pm \frac{1}{2} e^{-2\gamma t} \,. \qquad (48)$$

**Paired commuting density matrices**   Paired commuting density matrices take the form (44), where in the present case $\bar{\Gamma}_k^\dagger, \bar{\Gamma}_k = \bar{\eta}_k^\dagger, \bar{\eta}_k$. Plugging in the time evolution of the latter, we check that the corresponding eigenvalues take the form

$$p_{k_1,k_2}^{\mathrm{C}(1,2,3,4)}(t) = \left( \frac{1}{2} + \frac{\varepsilon}{2} e^{-2\gamma t} \right) \left( \frac{1}{2} + \frac{\varepsilon'}{2} e^{-2\gamma t} \right) \,, \qquad \varepsilon, \varepsilon' = \pm 1 \,, \qquad (49)$$

irrespectively of the value of $\varphi_{k_1,k_2}$, as well as of the value of $k_1$ and $k_2$.

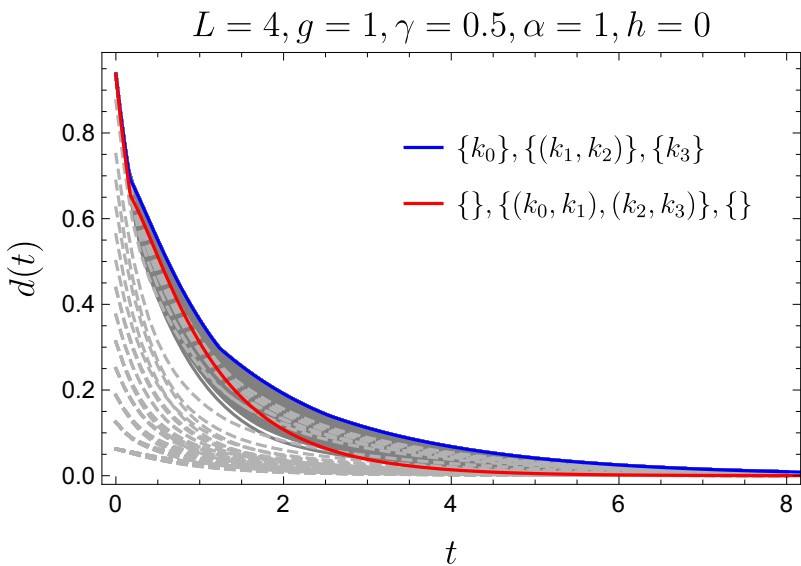

Figure 4: Evolution of the distance to equilibrium in the gain/loss model for $L = 4$. The dashed light gray lines correspond to randomly drawn initial density matrices with various levels of mixing, while the darker gray lines correspond to initial pure quantum states. The colored lines are analytical predictions for initial density matrices of the type (46). We recall the notation $k_j = \frac{2\pi j}{L}$ for the momenta, but emphasize that in the present case the results are insensitive to any permutation of the latter.

**Single-sector non-commuting density matrices** The case of single-sector non-commuting density matrices was already briefly discussed in the beginning of Section 4.2. These evolve as

$$\rho_k(t) = \frac{1}{2}\mathrm{id} + \left(p_k - \frac{1}{2}\right)\left[e^{-2\gamma t}\cos(2\theta_k)[\bar{\eta}_k^\dagger, \bar{\eta}_k] + e^{-\gamma t}\sin(2\theta_k)\left(e^{i(\varphi_k - \epsilon_k t)}\bar{\eta}_k^\dagger + e^{-i(\varphi_k - \epsilon_k t)}\bar{\eta}_k\right)\right],$$
(50)

and the maximal contribution to distance is obtained by setting $p_k = 1$ (or 0) and $\theta_k = \frac{\pi}{4}$, irrespectively of the value of $\varphi_k$. The associated eigenvalues then read :

$$p_k^{(1,2)}(t) = \frac{1}{2} \pm \frac{1}{2}e^{-\gamma t}.$$
(51)

Gathering (48), (49), (51), it is easy to see that the density matrix of the form (46) which maximizes the distance to equilibrium at all times corresponds to a product of one single-sector non-commuting density matrix, and commuting density matrices in all other sectors (given the similarity between (48) and (49), it does not matter which of those are paired and which are single-sector). On Figure 4, we represent the associated distance as a function of time for a system of finite size $L = 4$ (blue curve), and plot in comparison the distances computed from exact diagonalization starting from randomly drawn initial conditions, corresponding to either pure or mixed states. We also represent as a red curve the distance for a product of commuting (paired or single-sector) density matrices, which, following the same lines as above, corresponds to the fastest mixing (that is, minimizes the distance to equilibrium at time $t$) when restricting to pure quantum initial state of the form (46).

Several observations can be made from there. First, it is apparent that at any time the least mixed state is indeed of the form (46), with one single-sector fermionic density

matrix, and commuting matrices in other sectors. This observation continues to hold for larger systems sizes and we conjecture it to be true for generic $L$, however we limit ourselves to presenting results for a small system here, as for larger systems the Hilbert space of possible initial states becomes longer to explore, and the upper bound much harder to saturate from randomly drawn samples. In Section 5, we will study the large $L$ behaviour of the corresponding distance $d(t)$, with particular interest in whether a cutoff develops in this limit. Another observation is that, conversely, the lower bound for the distance (once restricted to starting from pure quantum states) is not given by a density matrix of the form (46) (red curve), which means that the states with fastest mixing may be non-factorizable.

## 4.4 The topological model

Let us now follow the exact same steps for the topological model.

**Single-sector commuting density matrices**  As has been discussed in Section 4.2, the time evolution of the single-sector commuting density matrices has the same form as for the gain/loss model. Once again, their maximal contribution to the distance is obtained by choosing $p_k = 1$ (or 0) in (41), with eigenvalues given by (48).

**Paired commuting density matrices**  Paired commuting density matrices take the form (44). Their time evolution is read off by rewriting the $\bar{\Gamma}_k, \bar{\Gamma}_k^\dagger$ in terms of the master modes using (28)-(30), and we find that as a function of time the corresponding eigenvalues take the form

$$p_{k_1,k_2}^{\mathrm{C}(1,2,3,4)}(t) = \left(\frac{1}{2} + \frac{\varepsilon}{2} f_{k_1,k_2}^+(t)\right)\left(\frac{1}{2} + \frac{\varepsilon'}{2} f_{k_1,k_2}^-(t)\right), \qquad \varepsilon, \varepsilon' = \pm 1, \qquad f_{k_1,k_2}^+(t) f_{k_1,k_2}^-(t) = e^{-4\gamma t}.$$
(52)

where the explicit form of $f_{k_1,k_2}^\pm(t)$ depends on $\varphi_{k_1,k_2}$. The corresponding contribution to the distance is maximized at a given time $t$ for a certain value $\varphi_{k_1,k_2}^*(t)$ of $\varphi_{k_1,k_2}$, and the associated $f_{k_1,k_2}^{*\pm}(t)$ will be given in Section 5 for the particular case of the Ising chain, $\alpha = 1, h = 0$.

**Single-sector non-commuting density matrices**  The generic form for a single-sector (non necessarily commuting) initial density matrix is

$$\rho_k(0) = \frac{1}{2}\mathbf{1} + \left(p_k - \frac{1}{2}\right)\left[\cos(2\theta_k)[\bar{\Gamma}_k^\dagger, \bar{\Gamma}_k] + \sin(2\theta_k)\left(e^{i\varphi_k}\bar{\Gamma}_k^\dagger + e^{-i\varphi_k}\bar{\Gamma}_k\right)\right].$$
(53)

As above we can compute the time evolution $\rho_k(t)$ by recasting $\bar{\Gamma}_k, \bar{\Gamma}_k^\dagger$ in terms of the master modes, and find the associated eigenvalues under the form

$$p_k^{(1,2)}(t) = \frac{1}{2} \pm \frac{1}{2} f_k(t),$$
(54)

where once again the function $f_k(t)$ depends on the parameters $\theta_k, p_k, \varphi_k$. The maximal distance at time $t$ is obtained by maximizing $|f_k(t)|$, which corresponds to a choice of parameters

$$p_k = 1, \qquad \theta_k = \frac{\pi}{4}, \qquad \varphi_k = \varphi_k^*(t) := \frac{\pi}{2} + \frac{i}{2}\tanh^{-1}\left(\frac{\sqrt{\epsilon_k^2 - \gamma^2}}{\epsilon_k}\coth(t\sqrt{\epsilon_k^2 - \gamma^2})\right).$$
(55)

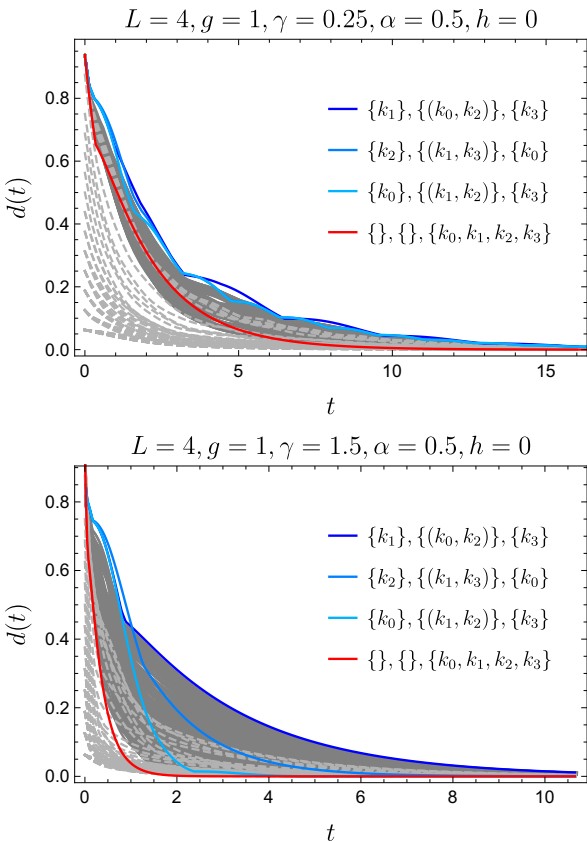

Figure 5: Evolution of the distance to equilibrium in the topological model for $L = 4$ and various choices of parameters. The dashed light gray lines correspond to random initial density matrices with various levels of mixing, while the darker gray lines correspond to initial pure quantum states. The colored lines are analytical predictions for initial density matrices of the type (46), where we recall the notation $k_j = \frac{2\pi j}{L}$.

The corresponding explicit expression of $f_k(t)$, which we denote $f_k^*(t)$, is too lengthy to be detailed here, but will be given in Section 5 for the particular case $\alpha = 1$.

As for the gain/loss model we now compare the distances built from (48), (52), (54) with numerical results from randomly drawn initial density matrices, see Figure 5. At difference with the gain/loss model, the eigenvalues (48), (52), (54) now depend on the momentum sectors, and differ between the paired or single sector commuting cases. The distance to equilibrium therefore depends on the repartition of commuting matrices between paired and single-sector, as well as on the associated distribution of momenta, the choice of pairings between those, etc... We do not attempt at a generic discussion here (in Section 5 this discussion will be simplified by restricting to the particular case $\alpha = 1, h = 0$, where the dependence in momentum vanishes), but restrict to plotting for $L = 4$ and different sets of parameters the distances associated with some relevant initial states, including the slowest and fastest mixing cases (blue and red curves, respectively). As can be observed from the blue curves in Figure 5, the maximal distance is attained at any time by a product of one non-commuting single sector density matrix, and commuting matrices in all other sectors, with the maximal of these being paired (which leaves out, for $L$ even, one commuting single sector). We also note, as attested by the various crossings between blue curves, that the repartition of momenta for which this maximal distance is

attained may vary over time.

Turning to the fastest mixing states (restricted to start from a pure quantum state), two regimes emerge, which seem to coincide with the regimes $\gamma/g > 2 - |h|$, $\gamma/g < 2 - |h|$ emerging from the band structure on Figure 2. In the "small dissipation" regime ($\gamma/g < 2 - |h|$, left panel on Figure 5) things seem to go similarly as in the gain-loss model, namely the fastest mixing state is not of the form (46) and therefore corresponds to a non-factorizable state. In contrast, in the "large dissipation" regime ($\gamma/g > 2 - |h|$, right panel on Figure 5), fastest mixing does seem to be attained for a product of single sector commuting density matrices, and we further observe a separation of timescales between the fastest and slowest mixing states. This separation of timescales will be studied in more detail in the next section.

## 5 Study of the mixing times and cutoffs

In the previous section we have constructed a family of factorized density matrices (46), which, despite being a very restricted subset among all the possible initial states, do achieve at all times the slowest mixing (that is, they maximize at all times the distance (37) to equilibrium), as well as, in some regimes of parameters, the fastest mixing. We will now exploit the analytical formula (47) for the associated distance to equilibrium in order to investigate the mixing properties and existence of cutoffs in the different regimes of our models. Then, we will turn in Sections 5.3 and 5.4 to other related aspects, namely the behaviour of other physical observables with respect to mixing, and the effect of the edge mode discussed in Section 3.3 when taking open boundary conditions in the topological model.

### 5.1 The gain-loss model

As we have seen in Section 4.3, the slowest mixing in the gain-loss model is obtained at all times starting from a density matrix of the form (46), with one non-commuting single sector density matrix and commuting matrices in all other sectors. Using (47), the corresponding distance to equilibrium as a function of time reads

$$d(t) = \frac{1}{2} \sum_{\varepsilon_1 = \pm, \dots \varepsilon_L = \pm} \left| \left( \frac{1}{2} + \varepsilon_1 \frac{e^{-\gamma t}}{2} \right) \prod_{k=2}^{L} \left( \frac{1}{2} + \varepsilon_k \frac{e^{-2\gamma t}}{2} \right) - \frac{1}{2^L} \right|, \tag{56}$$

(where once again the signs $\varepsilon_k = \pm$ have nothing to do with the energies $\epsilon_k$), and can be reinterpreted as the distance to equilibrium for a classical random walk on an anisotropic hypercube, with jump rate $\gamma$ in one direction and $2\gamma$ in the other directions.

The large $L$ asymptotics of (56) can be tackled analogously as in the isotropic case, and we show in Appendix B that for $L$ large, $t$ large,

$$d(t) \simeq \frac{1}{2} \left( \left( 1 - \frac{e^{-\gamma t}}{2} \right) \text{erf} \left( \frac{\sqrt{L} e^{-2\gamma t}}{\sqrt{8}} - \frac{e^{\gamma t}}{\sqrt{2L}} \right) + \left( 1 + \frac{e^{-\gamma t}}{2} \right) \text{erf} \left( \frac{\sqrt{L} e^{-2\gamma t}}{\sqrt{8}} + \frac{e^{\gamma t}}{\sqrt{2L}} \right) \right), \tag{57}$$

where erf is the Gauss error function. The function (57) develops a cutoff for

$$t_{\text{mix}}(L) = \frac{\ln L}{4\gamma} = \frac{t_{\text{rel}}}{4} \ln L, \tag{58}$$

where the relaxation time $t_{\rm rel}$, defined in (23), is the inverse of the spectral gap. The asymptotic profile of the cutoff around this time is

$$d\left(\frac{\ln L}{4\gamma} + s\right) = \mathrm{erf}\left(\frac{e^{-2\gamma s}}{\sqrt{8}}\right).$$

(59)

Several remarks are in order : first, the profile (59) is formally the same as for the classical hypercube random walk, and in particular retains a finite width when $L \to \infty$. Second, it is independent of the coherent coupling strength $g$. Nevertheless, a striking difference with the classical case is that all timescales, and in particular the mixing time (58), are divided by two as a consequence of the constraint imposed on initial states by the fermionic nature of the problem, and which can be viewed as a kind of exclusion constraint (see section 4.2).

## 5.2 The topological model

For the topological model, we have seen in Section 4.4 that the least mixed state is obtained at any time from a factorized matrix of the form (46) with one non-commuting factor and paired commuting density matrices in other sectors, plus one residual single sector commuting matrix in the case where $L$ is even. Furthermore, the way momenta $\{k_0, \ldots k_{L-1}\}$ should be distributed between all these factors in order to maximize the distance (47) may generically depend on time in a complicated fashion as attested by the multiple crossings between the blue curves on Figure 5.

Here, we will specify to the particular case $\alpha = 1$, $h = 0$, that is that of the Ising chain in the absence of an external magnetic field, which brings the simplification that the energies $\epsilon_k$ in (14) and hence the Liouvillian eigenvalues $\beta_k^{\pm}$ do not depend on $k$, namely $\epsilon_k = 2g$ for all $k$. In practice this means that all the blue curves on each panel of Fig. 5 collapse into a single one, for which we will be able to compute the large $L$ asymptotics. The distance reads from (47)

$$d(t) = \frac{1}{2} \sum_{\varepsilon_1, \ldots \varepsilon_L = \pm} \left| \left(\frac{1}{2} + \frac{\varepsilon_1}{2} f_k^*(t)\right) \left(\frac{1}{2} + \frac{\varepsilon_2}{2} e^{-2\gamma t}\right) \prod_{a=1}^{\frac{L}{2}-1} \left(\frac{1}{2} + \frac{\varepsilon_{2a+1}}{2} f_{k_{i_a}, k_{j_a}}^{*+}(t)\right) \left(\frac{1}{2} + \frac{\varepsilon_{2a+2}}{2} f_{k_{i_a}, k_{j_a}}^{*-}(t)\right) - \frac{1}{2^L} \right|$$

(60)

where the functions $f_k^*(t)$ and $f_{k',k''}^{*\pm}(t)$ are those appearing in (54) and (52), taken for the worst-choice values of the initial state parameters, and which we here find to be

$$f_k^*(t) = e^{-t\gamma} \left( \sqrt{1 + \left(\frac{\gamma \sinh(t\sqrt{\gamma^2 - 4g^2})}{\sqrt{\gamma^2 - 4g^2}}\right)^2} + \frac{\gamma \sinh(t\sqrt{\gamma^2 - 4g^2})}{\sqrt{\gamma^2 - 4g^2}} \right),$$

(61)

$$f_{k',k''}^{*+}(t) = (f_k^*(t))^2, \qquad f_{k',k''}^{*-}(t) = \frac{e^{-4\gamma t}}{(f_k^*(t))^2}.$$

(62)

As for the gain-loss model, we can reinterpret (47) as the distance for a classical random walk on an anisotropic hypercube, with the additional difference that the jumps are not exactly Poisson processes anymore, as the functions (61), (62) are not purely exponentially decaying, but only become so in the late time limit.

In Appendix B we show from there that in the large $L$ limit all of these distances exhibit a cutoff phenomenon at times $\propto \ln L$, and that the asymptotic expressions the cutoff profiles can be expressed in terms of the Gauss error function erf. In order to describe these profiles in more detail we now distinguish between the small disipation ($\gamma < 2g$)

and large dissipation regimes ($\gamma > 2g$), recalling that the spectral gap (35), becomes for $\alpha = 1, h = 0$,

$$\bar{\lambda} = \begin{cases} \gamma & |g| \geq \frac{\gamma}{2}, \\ \gamma - \sqrt{\gamma^2 - 4g^2} & |g| \leq \frac{\gamma}{2}. \end{cases} \tag{63}$$

We will also be interested in the asymptotics of the "fast mixing" red curves of Figure 5, which is given in both regimes by

$$d_{\text{fast}}(t) = \frac{1}{2} \sum_{p=0}^{L} \binom{L}{p} \left| \left( \frac{1}{2} + \frac{e^{-2\gamma t}}{2} \right)^p \left( \frac{1}{2} - \frac{e^{-2\gamma t}}{2} \right)^{L-p} - \frac{1}{2^L} \right|. \tag{64}$$

This is exactly the distance to equilibrium for a classical random walk on the isotropic hypercube, and therefore develops a cutoff at times

$$t_{\text{mix}}^{(\text{fast})}(L) = \frac{\ln L}{4\gamma}. \tag{65}$$

As has been discussed in section 4.4 these red curves do not necessarily correspond to the fastest mixing state (only in the large dissipation regime might it be the case), but looking at Fig. 5 it seems reasonable to expect that in both regimes these give a good indication of the spreading of mixing times for all possible initial states (conditioned to be pure quantum states at $t = 0$).

### 5.2.1 Small dissipation regime ($\gamma < 2g$)

In the small dissipation regime, we find following Appendix B the asymptotics

$$d(t) \simeq \text{erf} \left( \frac{e^{-2\gamma \left(t - \frac{\ln L}{4\gamma}\right)}}{2\sqrt{2}} \sqrt{1 + 8 \left( \frac{\gamma \sin(2t\sqrt{4g^2 - \gamma^2})}{2\sqrt{4g^2 - \gamma^2}} \right)^2 + 8 \left( \frac{\gamma \sin(2t\sqrt{4g^2 - \gamma^2})}{2\sqrt{4g^2 - \gamma^2}} \right)^4} \right), \tag{66}$$

which develops a cutoff for

$$t_{\text{mix}}(L) = \frac{\ln L}{4\gamma} = \frac{t_{\text{rel}}}{4} \ln L. \tag{67}$$

The corresponding distance, as well as the "fast mixing" distances are plotted on the left panel of Figure 6 for various sizes.

The situation here is quite similar to that of the gain/loss model : the slowest and fast mixing curves develop a cutoff at the same value mixing time (67), which is half that of the classical problem. Assuming, as seemed reasonable from Figure 5, that the distance for generic (pure) initial states remains "close enough" to these two cases, we may conclude that in this regime all initial states develop a cutoff at time (67).

### 5.2.2 Large dissipation regime ($\gamma > 2g$)

In the large dissipation regime, the asymptotics of the functions (61), (62) further simplifies to

$$f_k^*(t) \sim \frac{\gamma}{\sqrt{\gamma^2 - 4g^2}} e^{-t\left(\gamma - \sqrt{\gamma^2 - 4g^2}\right)} \tag{68}$$

$$f_{k',k''}^{*\pm}(t) \sim \left( \frac{\gamma^2}{\gamma^2 - 4g^2} \right)^{\pm 1} e^{-2t\left(\gamma \mp \sqrt{\gamma^2 - 4g^2}\right)}. \tag{69}$$

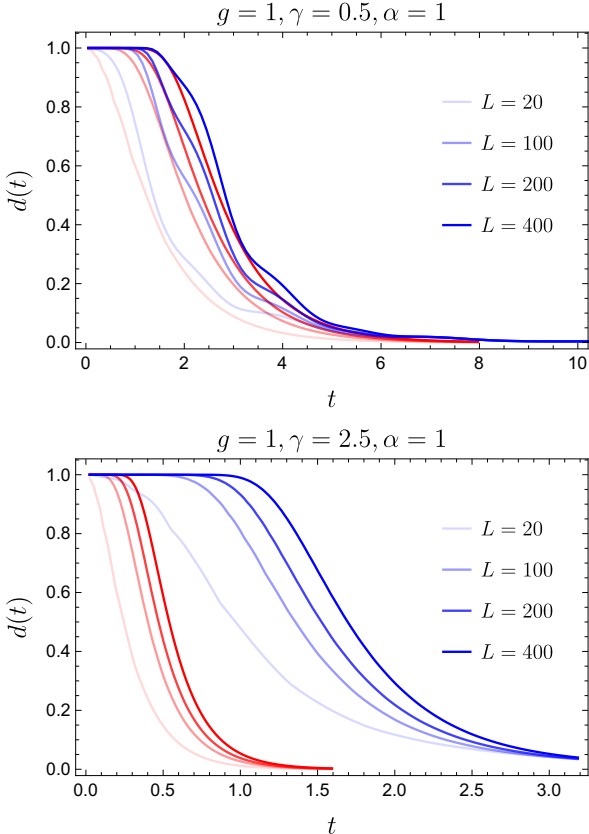

Figure 6: Distance to equilibrium in the topological model for the Ising chain in zero magnetic field ($\alpha = 1, h = 0$) starting from two different classes of initial states, in the regimes $\gamma > 2g$, $\gamma < 2g$ and for different sizes $L$. The blue curves correspond to the least mixed state at all time ("worst choice" initial conditions), while the red curves correspond to a product of single-sector commuting matrices, which for $\gamma > 2g$ seems to be the fastest-mixing state.

Following Appendix B, we check that the distance (60) develops a cutoff at

$$t_{\text{mix}}(L) = \frac{\ln L}{4(\gamma - \sqrt{\gamma^2 - 4g^2})} = \frac{t_{\text{rel}}}{4} \ln L \,, \tag{70}$$

and the cutoff profile is given by

$$d\left(\frac{\ln L}{4(\gamma - \sqrt{\gamma^2 - 4g^2})} + s\right) = \text{erf}\left(\frac{\gamma^2}{4(\gamma^2 - 4g^2)} e^{-2(\gamma - \sqrt{\gamma^2 - 4g^2})s}\right) \,. \tag{71}$$

An interesting feature appearing here, illustrated on the right-hand panel of Figure 6, is the separation of timescales between the slowest mixing states and the "fast mixing" states (which, as concluded from Figure 5, are likely to be the fastest mixing states in this regime). In other terms, the time required to reach equilibrium strongly depends on the initial state in this regime. Let us point that the fact that both the fastest and slowest mixing states show a cutoff phenomenon does not imply a cutoff phenomenon for any initial state. Rather, we conclude from the above that any initial state (conditioned to be a pure quantum state at $t = 0$) should display a weaker version known as *pre-cutoff* [10,16] , characterized by the two sets of timescales $t_{\text{mix}}(L)$ (eq. (65)) and $t_{\text{mix}}^{(\text{fast})}(L)$ (eq. (70)) and the fact that for any $\epsilon > 0$,

$$||\rho\left((1 - \epsilon)t_{\text{mix}}^{\text{fast}}(L)\right) - \rho_\infty|| \xrightarrow[L \to \infty]{} 1 \tag{72}$$

$$||\rho\left((1 + \epsilon)t_{\text{mix}}(L)\right) - \rho_\infty|| \xrightarrow[L \to \infty]{} 0 \,. \tag{73}$$

## 5.3 Other physical observables

In the previous sections we have observed that both the gain/loss and topological models exhibit a cutoff phenomenon, as defined by the trace-norm distance to equilibrium of the slowest-mixing state. More generally, it seems that the distance from an arbitrary initial state satisfies a pre-cutoff with two timescales $t_{\text{mix}}^{(\text{fast})}(L)$ and $t_{\text{mix}}(L)$. In the gain/loss model and the weak-dissipation regime of the topological model these two timescales coincide, so there is in fact a cutoff from any initial state.

A natural question at this stage, is whether the notion of cutoff extends to other physical observables, for instance to the growth of the von Neumann entropy $S(t) = -\text{Tr}(\rho(t) \ln \rho(t))$ or to the equilibration of local observables. Let us start with the entropy. Starting from initial states of the form (46), it is easy to see that it can be decomposed at all times as a sum over individual (pairs of) sector contributions,

$$S(t) = \left(-\sum_{c \in \{1,2\}} p_k^{(c)} \ln p_k^{(c)}\right) - \sum_{a=1}^{m} \sum_{d_a \in \{1,2,3,4\}} p_{k_{i_a}, k_{j_a}}^{\text{C}(d_a)} \ln p_{k_{i_a}, k_{j_a}}^{\text{C}(d_a)} - \sum_{b=1}^{m} \sum_{e_b \in \{1,2\}} p_{k_{l_b}}^{\text{C}(e_b)} \ln p_{k_{l_b}}^{\text{C}(e_b)} \,, \tag{74}$$

which, for both the slowest- and fast- mixing states of the gain/loss model, converges for $L \to \infty$ to the asymptotic expression

$$\lim_{L \to \infty} \frac{S(t)}{L} = -\frac{1 + e^{-2\gamma t}}{2} \ln(1 + e^{-2\gamma t}) - \frac{1 - e^{-2\gamma t}}{2} \ln(1 - e^{-2\gamma t}) + \ln 2 \,. \tag{75}$$

We plot on Figure 7 the trace norm distance and the (rescaled) entropy for the fastest-mixing state as a function of time for various system sizes. While the former develops a

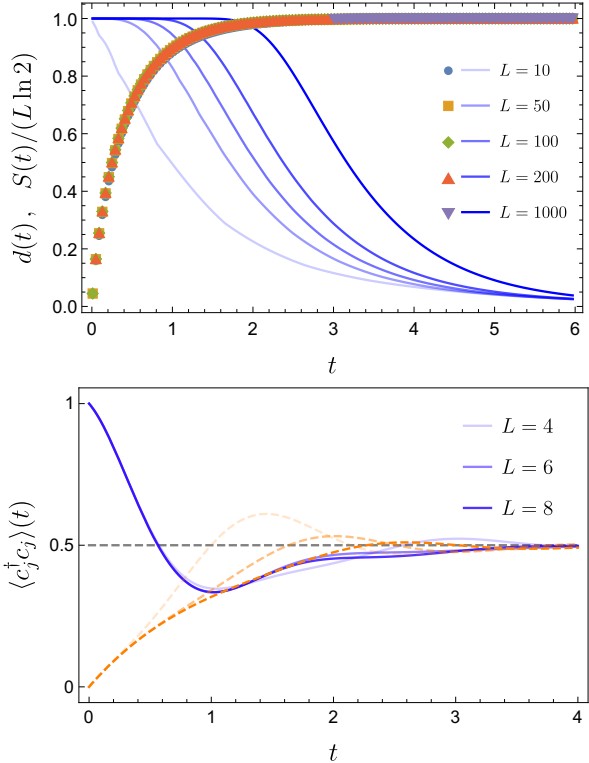

Figure 7: (Left) Evolution of the distance to equilbrium (blue curves) and the von Neumann entropy (colored dots) of the slowest-mixing state for various system sizes in the gain-loss model with $\gamma = 0.5$. (Right) Evolution of the fermion occupation number at site $L$ (blue curves) and in the middle of the chain (orange curves) for the gain/loss model with $\alpha = 0$, $h = 0$, $\gamma = 0.5$, starting from the state with one particle at site $L$ and zero elsewhere.

cutoff at times $t_{\mathrm{mix}}(L) \propto \ln L$, the latter relaxes exponentially with a timescale $1/2\gamma$ for any system size, and therefore is insensitive to the cutoff.

A similar conclusion can be made for local observables. This is in fact very easily seen in the classical case: starting from the configuration localized on site $\{0\}^L$ of the hypercube, a meaningful local observable is for instance the expectation value of the $i^{\mathrm{th}}$ coordinate. As can easily be checked this evolves in time as $\frac{1}{2} - \frac{e^{-\gamma t}}{2}$, so relaxes exponentially towards equilibrium with a timescale incommensurate with the mixing time. Now, there is little reason to expect that things should go differently in the quantum case. In order to illustrate this fact but keep the calculations at their simplest, let us consider the example of the gain/loss model with $\alpha = 0$, and look ath the evolution of the number of particles at site $j = L$. In this case the Bogoliubov fermions $\eta_k^\dagger, \eta_k$ coincide with the original momentum-space operators $c_k^\dagger, c_k$, and we have simply

$$\langle c_j^\dagger c_j \rangle(t) = \frac{1}{L} \sum_{k \neq k'} e^{-2\gamma t} \cos((\epsilon_k' - \epsilon_k)t + j(k' - k)) \langle c_k^\dagger c_{k'} \rangle(0) + \frac{1}{L} \sum_k \left( e^{-2\gamma t} \langle c_k^\dagger c_k \rangle(0) + \frac{1 - e^{-2\gamma t}}{2} \right) . \tag{76}$$

Starting for instance from the situation where there is one particle on site $j = L$ and none on the others (as we have argued earlier, this initial state, like all pure quantum initial states, is expected to develop a cutoff as $L \to \infty$), we have $\langle c_k^\dagger c_{k'} \rangle(0) = 1/L \ \forall k, k'$, which we can plug into (76) in order to obtain the various occupation numbers at time $t$. The results are plotted on the right-hand panel of Fig. 7 for $j = L$ and $j = L/2$. In all cases, the occupation numbers converge to their equilibrium values with a timescale $1/2\gamma$, which is the same as for the entropy and is incommensurate with the mixing time.

## 5.4 Effect of the edge mode

We finally move on to considering the effect of topological features on the mixing properties. For this sake we take the topological model with open boundary conditions, as studied in Section 3.3, where it was shown that there exists throughout the regime $|h| < 2$, $\alpha \neq 0$ a zero mode $\Psi$, commuting in the $L \to \infty$ limit with the action of the Liouvillian, and resulting in a twofold degeneracy of the Liouvillian spectrum between sectors $\mathcal{Z} = \pm 1$. While these features hold throughout the topological phase up to corrections exponentially small in $L$, they are exact at any size at the zero-field Ising point $\alpha = 1, h = 0$, where the action of $\Psi$ is simply given by (36). As a degeneracy of the spectrum means in particular closing of the spectral gap, we expect that taking open boundary conditions will have a drastic effect on the mixing properties. In fact, the whole discussion on mixing and cutoffs, including the definition (9) of the distance, is valid only in the case where the Liouvillian has one single non-degenerate steady state, which in the present case discards the particular point $\alpha = 1, h = 0$. For that case, the various initial states decay at large time towards linear combinations of the form $\frac{1}{2^L}(\mathrm{id} + f\sigma_L^x)$, $-1 \leq f \leq 1$, and the distance as defined in (9) may take any value between 0 and 1/2 in the $t \to \infty$ limit. Now, in the rest of the topological phase, the spectral gap is strictly speaking $> 0$ for any system size $L$, and any initial state will eventually decay towards the unique steady state $\rho_\infty = \frac{1}{2^L}\mathrm{id}$. The associated timescale is however now exponentially large in the system size, and cancels the cutoff effect observed for periodic boundary conditions. This is illustrated on Figure 8, where we compare the distance from randomly drawn initial states for open and periodic boundary conditions (we also plot in comparison the distances for the gain/loss model with open boundary conditions, which as should be expected does not show any drastic modification compared with the periodic case).

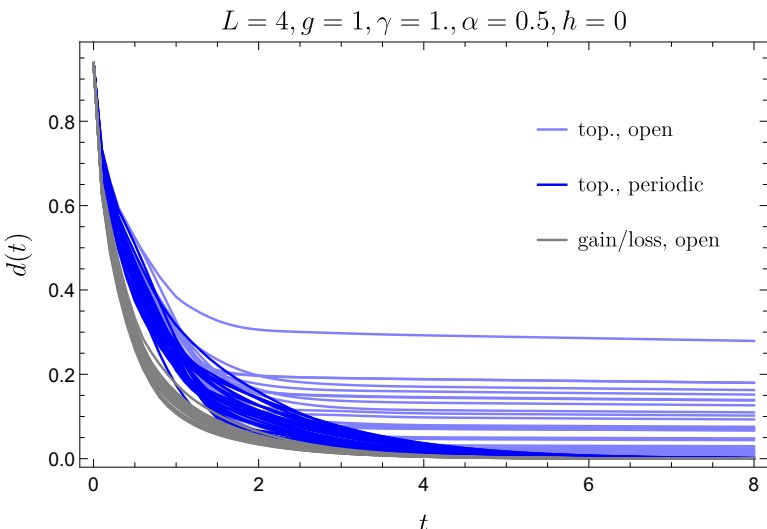

Figure 8: Comparison of the distance to equilibrium for the topological model for open and periodic boundary conditions, as well as the gain/loss model with open boundary conditions, on a system of size $L = 4$ and starting from randomly drawn initial states (restricted to be pure quantum states at time 0).

The long-lived edge mode also manifests itself at the level of "local" (in terms of spins, not of fermions) physical observables. The consequences of edge zero modes on the out-of equilibrium physics have been thoroughly studied in the past litterature, for closed Hamiltonian dynamics [29] but also in a dissipative (albeit quite different from the one considered here) setup [30, 31]. Here we illustrate these effects by comparing the time evolution of the expectation value $\langle \sigma_L^x \rangle$ for open and periodic boundary conditions, see Figure 9. While in the periodic case $\langle \sigma_L^x \rangle = \langle \sigma_1^x \rangle$ quickly relaxes to its equilibrium value, in the open case relaxation takes a much longer time, exponentially increasing with the system size. This is a result of $\sigma_L^x$ being "almost conserved" by the Liouvillian evolution, and we indeed check in comparison a much quicker relaxation at the other end of the chain, for the expectation value $\langle \sigma_1^x \rangle$.

# 6    Conclusions

In this work we have examined the mixing properties of open quantum fermionic systems whose time evolution is governed by the Lindblad equation, and in particular the quantum counterpart of the so-called cutoff phenomenon, a well-studied aspect of classical Markov chains. A general conclusion of our analysis is that the notion of cutoff generally carries over to the quantum framework (see also [16]): considering two free-fermionic models which in the classical limit reduce to the well-studied hypercube random walk [10, 12], we showed that the "worst-choice" distance to equilibrium, defined by maximizing over all possible initial conditions, develops a cutoff at a time $t_{\text{mix}}(L) \propto \ln L$ as $L \to \infty$. Going further, we explored the initial-state dependence of the mixing properties, and provided some evidence that a cutoff, or pre-cutoff (depending on the model and regime) should exist for arbitrary initial states.

With respect to the classical case, the quantum realm however reveals a number of new and potentially interesting features. A first aspect, well illustrated by the gain/loss

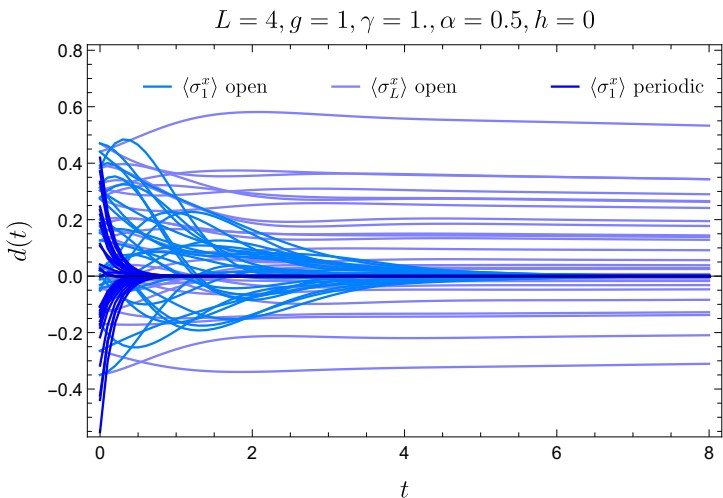

Figure 9: Evolution of the "local" observables $\langle \sigma_1^x \rangle$ and $\langle \sigma_L^x \rangle$ for the topological model with open and periodic boundary conditions, starting from randomly drawn initial states (restricted to be pure quantum states at time 0).

model, is the halving of the mixing times with respect to the classical problem, as a result of some "exclusion constraints" due to the fermionic nature of our models. This is in contrast, in particular, with the case of product quantum channels considered in [16], where, under some assumptions, the mixing time was related to the relaxation time through $t_{\mathrm{mix}}(L) = \frac{t_{\mathrm{rel}}}{2} \ln L$ (here, conversely, $t_{\mathrm{mix}}(L) = \frac{t_{\mathrm{rel}}}{4} \ln L$). Another aspect is the separation of timescales in some regimes between the slowest-mixing and fastest-mixing states. Finally, we have seen in the so-called "topological model" how the presence of an edge zero mode dramatically alters the mixing properties when open boundary conditions are taken, a phenomenon we know of no counterpart in the classical context (let us warn however, that the classical limit of open boundary conditions in the quantum chain does not correspond to open boundary conditions on the classical hypercube).

There are several interesting directions to go from there. The most natural next step would be to investigate the mixing properties of systems with a non-trivial steady state, for instance one with long-range order, entanglement [1], or current-carrying [3, 33, 34]. Another possible direction would be to study whether the present analysis, and in particular the existence of cutoffs, could be extended to non-markovian dynamics such as that governing the evolution of subsystem density matrices in isolated quantum systems after a quantum quench [35, 36], or in random quantum circuits [37] (see also [38]). A second direction concerns the relation with other physical observables : even though we have observed in Section 5.3 that the most natural local observables as well as the von Neumann entropy are insensitive to the presence of a cutoff, namely they do not develop a sharp jump as the trace-norm distance to equilibrium does at the mixing times $t_{\mathrm{mix}}(L)$, it remains an intriguing question whether the cutoff phenomenon might transpire into other physical quantities. On a more technical level, it would of course be interesting to move on to interacting systems, for instance using the mapping of some interacting Liouvillians onto Bethe ansatz solvable quantum Hamiltonians [39–42]. The study of mixing in this framework however seems to be a challenging issue. In fact, even in classical Markov chains which can be mapped onto quantum integrable systems (for instance the symmetric or assymetric exclusion processes [43–45]), relating mixing properties to the Bethe ansatz language seems to be a difficult task.

## Acknowledgments

I thank Lorenzo Piroli and Tomaž Prosen for very useful comments on the manuscript, as well as Siddarth Parameswaran for discussions.

## A  Random walk on the hypercube

Here we review well-known results on the random walk on the hypercube with many dimensions, as presented (up to changes in notations) in [12].

Consider a continuous time random walk on the hypercube $\{0,1\}^L$, parametrized by the $L$-uples $\mathbf{x} = (x_1, \ldots, x_L)$, where each $x_i \in \{0,1\}$. Starting in the situation where all $x_i = 0$, the walk undergoes neares-neighbour jumps at rate $L\gamma$. In other terms, every component $x_i$ changes value at rate $\gamma$, and its probability of being in state 1 at time $t$ is $\mathbb{P}(x_i = 1) = \frac{1}{2}(1 - e^{-\gamma t})$. At time $t$, the probability of being at a position $\mathbf{x}$ is therefore

$$\mathbb{P}(\mathbf{x}) = \frac{1}{2^L}(1 - e^{-\gamma t})^{|\mathbf{x}|}(1 + e^{-\gamma t})^{L-|\mathbf{x}|}, \tag{77}$$

where $|\mathbf{x}| = x_1 + \ldots + x_L$.

The stationary distribution, reached as $t \to \infty$, is the uniform distribution $\pi$ where every $\mathbf{x}$ has probability $\pi(\mathbf{x}) = 1/2^L$, and one is interested in the evolution of the total variation distance

$$d(t) = \frac{1}{2}\sum_{\mathbf{x}} |\mathbb{P}(\mathbf{x}) - \pi(\mathbf{x})| = \sum_{\mathbf{x}|\mathbb{P}(\mathbf{x}) \geq 1/2^L} (\mathbb{P}(\mathbf{x}) - \pi(\mathbf{x})). \tag{78}$$

Comparing with (77) shows that the condition $\mathbb{P}(\mathbf{x}) \geq 1/2^L$ is equivalent to imposing

$$|\mathbf{x}| \leq |\mathbf{x}|_{\max} = L\frac{\ln(1 + e^{-\gamma t})}{\ln \frac{1 + e^{-\gamma t}}{1 - e^{-\gamma t}}}. \tag{79}$$

When $N$ is large, $|\mathbf{x}|$, which is the sum of $L$ binomial variables, becomes a normal distribution of mean $\mu = L\frac{1 - e^{-\gamma t}}{2}$ and variance $\sigma^2 = L\frac{1 - e^{-2\gamma t}}{4}$. Similarly, under the distribution $\pi(\mathbf{x})$, the distribution of $|\mathbf{x}|$ is a normal distribution of mean $\frac{L}{2}$ and of variance $\frac{L}{4}$. We may therefore write

$$d(t) = \sum_{|\mathbf{x}| \leq |\mathbf{x}|_{\max}} (\mathbb{P}(\mathbf{x}) - \pi(\mathbf{x})) = \Phi_{\mu,\sigma}(|\mathbf{x}|_{\max}) - \Phi_{\frac{N}{2}, \frac{L}{4}}(|\mathbf{x}|_{\max}), \tag{80}$$

where $\Phi_{\mu,\sigma}$ is the cumulative distribution function of the normal law with parameters $(\mu, \sigma)$, that is,

$$\Phi_{\mu,\sigma}(z) = \frac{1}{\sigma\sqrt{2\pi}}\int_{-\infty}^{z} e^{-\frac{1}{2}\left(\frac{y-\mu}{\sigma}\right)^2} dy = \frac{1}{2} + \frac{1}{2}\text{erf}\left(\frac{z - \mu}{\sigma\sqrt{2}}\right). \tag{81}$$

We check from there that the distance $d(t)$ jumps from 1 to 0 at a time with increases logarithmically with $L$ (see Figure 1): more precisely, expanding to second order in $e^{-\gamma t}$, we obtain

$$d(t) = \text{erf}\left(\frac{e^{-\left(\gamma t - \frac{1}{2}\ln L\right)}}{\sqrt{8}}\right) + o(1) \tag{82}$$

# B    Asymptotic expressions for the various distances

Our goal here is to use techniques similar to those presented in Appendix A to derive the asymptotic behaviour of the distance $d(t)$ for "anisotropic" random walks as appearing in the main text.

We consider a random walk on the hypercube $\{0,1\}^L$, where now the different components change value at different rates. In fact, these are not necessarily Poisson processes, and we define the probabilities $\mathbb{P}(x_i = 1) = \frac{1}{2}(1 - f_i(t))$, where $f_i(t)$ are monotonously decreasing from 1 to 0 as $t$ grows from 0 to $\infty$, so that the stationary distribution $\pi$ is the uniform one. In practice we will consider the case of interest in Section 5.2, which is the case where $L$ is even, and

$$f_3(t) = f_4(t) = \ldots = f_{\frac{L}{2}+1}(t) := f_+(t) \tag{83}$$

$$f_{\frac{L}{2}+2}(t) = f_{\frac{L}{2}+3}(t) = \ldots = f_L(t) := f_-(t) \,. \tag{84}$$

Accordingly, we decompose $|\mathbf{x}| = x_1 + x_2 + |\mathbf{x}_+| + |\mathbf{x}_-|$. The probability of a given configuration at time $t$ reads

$$\mathbb{P}(\mathbf{x}) = \frac{1}{2^L} \prod_{a=1,2} (1 - f_a)^{x_a} (1 + f_a)^{1-x_a} \prod_{\epsilon=\pm} (1 - f_\epsilon)^{|\mathbf{x}_\epsilon|} (1 + f_\epsilon)^{\frac{L-2}{2} - |\mathbf{x}_\epsilon|} \,. \tag{85}$$

Proceeding as in Section A, computing the distance $d(t)$ implies a restricted summation over configurations with $\mathbb{P}(\mathbf{x}) \geq \frac{1}{2^L}$, which corresponds to

$$y \leq y_{\max}(x_1, x_2) \,, \tag{86}$$

where we have defined

$$y := |\mathbf{x}_+| \ln \frac{1 + f_+}{1 - f_+} + |\mathbf{x}_-| \ln \frac{1 + f_-}{1 - f_-} \tag{87}$$

$$y_{\max}(x_1, x_2) := \frac{L-2}{2} \left[ \ln(1 + f_+) + \ln(1 + f_-) \right] + \sum_{a=1,2} \ln((1 - f_a)^{x_a} (1 + f_a)^{1-x_a}) \,. \tag{88}$$

By virtue of the central limit theorem, when $L$ is large, $y$ is described by a normal law of parameters

$$\mu = \frac{L-2}{2} \left( \ln\left(\frac{1+f_+}{1-f_+}\right) \frac{1-f_+}{2} + \ln\left(\frac{1+f_-}{1-f_-}\right) \frac{1-f_-}{2} \right) \tag{89}$$

$$\sigma^2 = \frac{L-2}{2} \left( \left( \ln \frac{1+f_+}{1-f_+} \right)^2 \frac{1-f_+^2}{4} + \left( \ln \frac{1+f_-}{1-f_-} \right)^2 \frac{1-f_-^2}{4} \right) \,. \tag{90}$$

Similarly, under the uniform distribution $\pi(\mathbf{x})$, y is distributed under normal distribution of parameters

$$\mu_0 = \frac{L-2}{2} \left( \left( \ln \frac{1+f_+}{1-f_+} \right) \frac{1-f_+^2}{4} + \left( \ln \frac{1+f_-}{1-f_-} \right) \right) \tag{91}$$

$$\sigma_0^2 = \frac{L-2}{4} \left( \left( \ln \frac{1+f_+}{1-f_+} \right)^2 + \left( \ln \frac{1+f_-}{1-f_-} \right)^2 \right) \,. \tag{92}$$

Decomposing the distance as

$$d(t) = \sum_{x_1,x_2=0,1} \sum_{y \leq y_{\max}(x_1,x_2)} \left( p(x_1, x_2) \mathbb{P}(\mathbf{x}_+, \mathbf{x}_-) - \frac{1}{2^L} \right) \,, \tag{93}$$

we therefore obtain

$$d(t) = \sum_{x_1,x_2=0,1} \left[ p(x_1,x_2)\Phi_{\mu,\sigma}(y_{\max}(x_1,x_2)) - \frac{1}{4}\Phi_{\mu_0,\sigma_0}(y_{\max}(x_1,x_2)) \right] , \qquad (94)$$

where $\Phi_{\mu,\sigma}$ was defined in (81).

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
