# Peer review of "Mixing times and cutoffs in open quadratic fermionic systems"

_SciPost Physics_

## Round 1 · Referee Report · Anonymous · 2020-7-8

Strengths

1. Cutoff phenomena are interesting properties of certain Markov processes, so studying them for quantum systems is a natural and interesting question.
2. The analysis is straightforward but has some nice features, like being able to construct the slowest-mixing states.
3. The discussion is generally clear and the data on the heterogeneity of mixing properties is nice.

Weaknesses

1. At some level the system is just a bunch of decoupled two-level systems, so the existence of a cutoff is not surprising.
2. The interest of the topological model is that it has a phase transition. However, in Sec. 5 I did not see a useful discussion of what happens as one approaches this phase transition. Eq. (60) is cut off, which makes it hard to reconstruct the logic around here. In particular, since the spectral gap vanishes at the topological transition, one would think it should be simple to construct slowly mixing states at that point, but everything seems to decay as exp(-gamma t). I don't follow why this is happening.

Report

I think there is enough new content to merit publishing.

Requested changes

1. Fix the discussion of mixing in the topological model and discuss what happens near the topological transition.

  • validity: high
  • significance: good
  • originality: good
  • clarity: high
  • formatting: below threshold
  • grammar: excellent

Author:  Eric Vernier  on 2020-07-29  [id 909]

(in reply to Report 1 on 2020-07-08)

I thank the referee for his/her insightful comments on the manuscript. A new version of the manuscript has been prepared, where I hope to have addressed all the points raised in the above report. Please find below a few comments and explanations :

1- Regarding the fact that : "At some level the system is just a bunch of decoupled two-level systems, so the existence of a cutoff is not surprising." Indeed, that product chains lead to a cutoff is a well-known fact for classical Markov chains, and was proven under some assumptions for bosonic models in a paper by Kastoryano et al (Ref. 16). However in the fermionic case, it was not obvious (at least to me) from the start how a product structure might emerge from combinations of "anticommuting sectors". As a matter of fact the work Ref. 19 did prove bounds for mixing times, but not the existence of a cutoff for decoupled fermionic systems. In conclusion I agree that the emergence of a cutoff is probably not so surprising, but one of the main points of the present work was to unveil the mechanisms leading to it in fermionic models, and, most importantly, the dependence in the initial state.

2- Regarding the topological model : In fact, my main interest in this model is the topological phase itself, and in particular the two following features : (i) the existence of two regimes, one similar to the gain/loss case, the other of a "quantum Zeno" type, where increasing the dissipation results in diminishing the gap (ii) the existence of edge zero modes in the case of open boundary conditions, with interesting consequences on the mixing properties. For this reason, in Section 5.2 I have specified to $h=0$, well-inside the topological phase. I agree with the referee that taking larger values of $h$ and, in particular, approaching the critical line $h=2$ would lead to a closing of the gap, however this was not central to the present discussion. In a new version of the manuscript (which will be submitted in the next refereeing round), the discussion on the phase diagram of the topological model has been improved, and the regimes of interest better specified.

---

## Round 1 · Referee Report · Anonymous · 2020-9-14

Report

A so-called cutoff (rapid mixing) phenomenon was discovered and has been well explored in classical systems. This interesting phenomenon was shown to carry over to some quantum systems by Kastoryano, Reeb and Wolf [16], but has evidently received little attention in that context. Here the cutoff phenomenon is further investigated in quantum systems, namely, free-Fermion chains with two different types of couplings to an external environment. With a combination of analytical and numerical techniques, it is shown that these models can indeed develop a cutoff. The analyses seem to be very careful, and the presentation is clear. Several interesting directions for future work are sketched. In my opinion, this paper is suitable for publication in SciPost.

The formatting of Eq (60) needs to be fixed.

  • validity: -
  • significance: -
  • originality: -
  • clarity: -
  • formatting: -
  • grammar: -

Author:  Eric Vernier  on 2020-09-18  [id 977]

(in reply to Report 2 on 2020-09-14)

I thank the referee for his/her comments.
A new version of the manuscript has now been uploaded, where among other additions the formatting of Eq. (60) has been fixed.

---

## Editorial Decision

resubmitted